# *Lactobacillus plantarum* MA2 Ameliorates Methionine and Choline-Deficient Diet Induced Non-Alcoholic Fatty Liver Disease in Rats by Improving the Intestinal Microecology and Mucosal Barrier

**DOI:** 10.3390/foods10123126

**Published:** 2021-12-16

**Authors:** Yanping Wang, Yang Zhang, Jingnan Yang, Haoran Li, Jinju Wang, Weitao Geng

**Affiliations:** College of Food Science and Engineering, Tianjin University of Science & Technology, Tianjin 300457, China; ypwang40@163.com (Y.W.); scarlett2211@163.com (Y.Z.); 15022621887@163.com (J.Y.); tustlhr2019@163.com (H.L.); bioorange@163.com (J.W.)

**Keywords:** non-alcoholic fatty liver disease, *L**. plantarum* MA2, intestinal microecology, mucosal barrier

## Abstract

Non-alcoholic fatty liver disease (NAFLD) has become a highly concerned health issue in modern society. Due to the attentions of probiotics in the prevention of NAFLD, it is necessary to further clarify their roles. In this study, the methionine and choline-deficient (MCD) diet induced NAFLD rats model were constructed and treated with strain *L*. *plantarum* MA2 by intragastric administration once a day at a dose of 1 × 10^8^ cfu/g.bw. After 56 days of the therapeutic intervention, the lipid metabolism and the liver pathological damage of the NAFLD rats were significantly improved. The content of total cholesterol (TC) and total triglyceride (TG) in serum were significantly lower than that in the NAFLD group (*p* < 0.05). Meanwhile, the intestinal mucosal barrier and the structure of intestinal microbiota were also improved. The villi length and the expression of claudin-1 was significantly higher than that in the NAFLD group (*p* < 0.05). Then, by detecting the content of LPS in the serum and the LPS-TLR4 pathway in the liver, we can conclude that Lactobacillus plantarum MA2 could reduce the LPS by regulating the gut microecology, thereby inhibit the activation of LPS-TLR4 and it downstream inflammatory signaling pathways. Therefore, our studies on rats showed that *L**. plantarum* MA2 has the potential application in the alleviation of NAFLD. Moreover, based on the application of the strain in food industry, this study is of great significance to the development of new therapeutic strategy for NAFLD.

## 1. Introduction

Non-alcoholic fatty liver disease (NAFLD) covers a wide range of liver diseases from hepatic steatosis to non-alcoholic steatohepatitis (NASH) [1]. In recent years, with the change of people’s lifestyle and diet structure, NAFLD has become the most widely distributed chronic liver disease in the world [2,3]. It is reported that the incidence of NAFLD in overweight and obese people is as high as 57% and 98% respectively. Due to the increasing incidence rate, NAFLD has become a highly concerned health issue in modern society [4].

NAFLD is a complex disease, which may be caused by many factors, including genetics, diet and intestinal microbiota [5,6,7,8]. In 1998, Day and James proposed the “two-hit” hypothesis to describe the pathogenesis of NAFLD [9]. According to the theory, the first hit came from lipid deposition in the liver. Subsequently, “the second hit” (such as lipopolysaccharide, LPS) triggered the development of hepatic steatosis to steatohepatitis [10], and caused liver cell death and liver inflammation [11,12], eventually leading to NASH and liver fibrosis [13,14]. Recently, a more comprehensive “multiple-hit hypothesis” has been proposed to summarize the complexity of NAFLD pathogenesis [15,16]. In this hypothesis, inflammatory factors from various tissues (especially adipose tissue and intestinal tissue) are involved in the activation of inflammation, the progression of fibrosis and tumorigenesis. Although the understanding of NAFLD has been improved, the progress of its clinical treatment is still limited [17]. Thus, so far, no drug has been approved for the treatment of NASH. An effective strategy for the treatment of NAFLD is to lose at least 5% of body weight in overweight and/or obese patients [5]. However, due to the poor compliance of patients, the treatment is difficult to be widely used in clinical treatment.

With the development of the concept of “gut liver axis”, the study of intestine-liver interaction is more and more in-depth. It is recognized that intestinal dysbacteriosis plays an important role in the pathogenesis of human liver diseases, especially in NAFLD and related metabolic disorders [6]. Therefore, it is promising to treat NAFLD by targeting the intestinal microbiota. Among them, probiotics, which can survive in the gut and improve intestinal health, have been paid more and more attention in the treatment and prevention of NAFLD. Probiotic therapy becomes a novel, safe and effective method, which can reverse the metabolic abnormalities observed in obesity, and is expected to become an alternative therapy for liver diseases such as NAFLD [6]. For example, studies have shown that oral supplementation of 10^10^ CFU probiotics (e.g., *Lactobacillus* and *Bifidobacterium*) for 12 weeks can reduce liver fat accumulation in high-fat diet (HFD) feeding Sprague–Dawley rats [18]. Dietary supplementation with 1 × 10^9^ CFU of *L*. *lactis subsp cremoris* ATCC 19257 three times per week for 16 weeks in female C57BL/6 mice on a high-fat, high-carbohydrate diet caused them to gain less weight, develop less liver fat and inflammation [19].

Therefore, to further clarify the role of probiotics in the intervention of NAFLD, the effect of a probiotic strain *L**. plantarum* MA2 on NAFLD and its action mechanism were investigated. In our previous study, the strain was proved to be able to colonize in the intestinal tract of mice and had a role in regulating the intestinal microbiota [20,21,22]. In this study, MCD-induced NAFLD rats were orally supplemented with the strain, and markers of lipid metabolism, liver damage, intestinal microbiota structure and intestinal barrier were examined. This study aimed to provide reference information for probiotics in treating NAFLD to form the novel intervention and treatment strategy. Moreover, based on the potential application of the strain *L**. plantarum* MA2 in food industry, this study is of great significance to the development of new therapeutic strategy for NAFLD.

## 2. Materials and Methods

### 2.1. Strain and Culture Conditions

*L**. plantarum* MA2 (CGMCC 12541) was isolated from fermented vegetables and stored in China General Microbiological Culture Collection Center (Beijing, China). Before the animal experiment, the *L**. plantarum* MA2 cells were collected by centrifuge and vacuum freeze-dried to obtain the powder of the strain with a count of 5 × 10^12^ CFU/g. Before use, physiological saline was used to prepare probiotic suspension with bacterial concentration of 10^9^ CFU/mL. The number of viable bacteria in the dried probiotic powder was determined according to the method described in the national food safety standard (food microbiological examination: lactic acid bacteria, GB_4789_35-20106).

### 2.2. Animal Treatment and Sample Collection

Animal experiments were carried out in the Animal Experiment Center of Tianjin University of Science and Technology under the permission of Tianjin experimental animal management office (Approval Number: TUST20191015) and in accordance with the “*Guide to the Management and Use of Experimental Animals*”. Male specific pathogen free (SPF) Sprague–Dawley rats (8 weeks old) weighing 200–230 g were selected for the experiment [23]. The environmental conditions of animal feeding were as follows: the humidity was 55 ± 5%, the temperature was 25 °C, and light/dark alternate for 12 h. After 7 days of adaptation, all the rats except the normal group were fed with methionine choline deficiency (MCD) diet for 8 weeks to induce NAFLD disease. The formula of MCD diet was as follows: L-amino acid 175.7 g/kg, sucrose 441.9 g/kg, corn starch 150.0 g/kg, dextran maltose 50.0 g/kg, cellulose 30.0 g/kg, corn oil 100.0 g/kg, sodium bicarbonate 7.4 g/kg, salt mixture 35.0 g/kg and vitamin mixture 10.0 g/kg. During the experiment, rats drank and ate freely.

In this experiment, the animals were divided into three groups. The animals in normal group (normal group, *n* = 8) were fed with normal diet and given sterile water by gavage. The animals in NAFLD model group (NAFLD group, *n* = 8) were fed with MCD diet and gavage with sterile water. The animals in *L*. *plantarum* MA2 treatment group (MA2 group, *n* = 8) were fed with MCD diet and gavaged with the probiotic suspension (10^9^ CFU of *L*. *plantarum* MA2 cells per mL) at amount of 1 × 10^8^ cfu/g.bw. The feeding, drinking, behavior, activity, mental state, hair condition and feces of the animals were observed every day, and the animals were weighed every week. After 24 h of the intragastric administration on Day 56, 3 fecal samples were randomly collected from each group and stored at −80 °C for high-throughput sequencing. At the end of the experiment (after 8 weeks), the animals were anesthetized with 10% chloral hydrate, and the blood was collected by heart puncture. Blood samples were stored at 4 °C for 1 h, then centrifuged at 3000× *g* for 10 min to separate serum. The serum samples were then stored at −80 °C for further use. The rats were dissected, and fresh liver and small intestine (ileum) tissues were collected. The weight, shape, size and color of the liver were observed and recorded, and the liver index (the ratio of liver weight to body weight) was calculated. Part of liver and ileum tissues were fixed in 10% formalin solution for more than 24 h, and the other part were stored at −80 °C for further detection.

### 2.3. Histological Observation and Evaluation

The liver and ileum samples (*n* = 6) and were randomly selected from each group to prepare paraffin sections, and the histomorphology was observed after hematoxylin-eosin (H&E) staining [24]. The samples for H&E staining were fixed with 4% paraformaldehyde for 24 h, paraffin-embedded liver specimens were systematically cut into sequential 4-µm-thick sections, and then stained with hematoxylin and eosin (H&E) The slices were imaged under a microscope (Leica DM3000, Wetzlar, Germany). The villi length was analyzed by using a NIS Elements software (version F3.2, Nikon Instruments, Melville, NY, USA).

The liver lesions were evaluated by the NAFLD disease score system designed by NASH pathological clinical research committee in 2005 [25]. The detailed information of the evaluation items and scores are listed as follows: hepatic steatosis (0–3 points according to the number of parenchymal cells/total cells are <5%, 5–33%, >33–66%, >66%, respectively), lobular inflammation (0–3 points, which are <2, 2–4 and >4 lesions, respectively), and hepatic cells showed balloon-like changes (0–2 points, no change, few balloon-like cells, many/prominent ballooning, respectively).

### 2.4. Detection of Biomarkers in Serum

The total cholesterol (TC), total triglyceride (TG), glutamic oxaloacetic transaminase (GOT/AST), glutamic pyruvic transaminase (GPT/ALT) and LPS in serums were detected by the corresponding kits. TC and TG in serum were determined by single reagent GPO-PAP method using total cholesterol assay kit (A111-1-1, Nanjing Jiancheng, China) and triglyceride assay kit (A110-1-1, Nanjing Jiancheng, China). GPT and GOT in serum were detected by Reiss method and microplate method, GTA Kit (C009-2-1, Nanjing Jiancheng, China) and aspartate aminotransferase assay kit (C010-2-1, Nanjing Jiancheng, China). The content of LPS in serum was determined by using a lipopolysaccharide assay kit (H255, Nanjing Jiancheng, China).

### 2.5. RNA Extraction and Real-Time PCR

The RNA extraction and real-time PCR was conducted according to the method described by Nasiri-Ansari et al. [26]. The total RNA in liver and small intestine was extracted with TRIzon reagent (CW0580, Cwbio, Beijing, China), and its concentration was determined by ultra-low volume spectrometer (BioDrop μLite. Biochrom Ltd. Shanghai, China). The RNA samples were immediately reverse transcribed into cDNA by a HiFi-MMLV cDNA Kit (CW0744, Cwbio, Beijing, China). The quantitative real-time PCR (RT qPCR) was conducted in a Real-Time PCR Detection System (CFX96, Bio-Rad, Hercules, CA, USA). BeyoFast™ SYBR Green qPCR Mix (2X) (D7260-5 mL, Beyotime, Shanghai, China) was used to amplify the target fragment. The primers used in this study and their target genes are listed in Table 1. The PCR procedure used in this study was as follows: pre denaturation (95 °C) for 2 min, followed by 40 cycles, denaturation (95 °C) 15 s, annealed (55 °C) 15 s and extension (72 °C) 30 s. At the end of the PCR procedure, the melting curve was observed at 65 °C to 95 °C to evaluate the specificity of the product. β-actin gene was used as housekeeping gene. The relative expression level of genes were normalized to that of β-actin and calculated according to the 2^−ΔΔCT^ approach.

### 2.6. Detection of Protein Expression

Detection of protein expression were used according to the method described by Zhuge et al. [27]. Total protein was extracted from liver tissue and quantified by the total protein assay kit (with standard: BCA method) (A045-4-2, Nanjing Jiancheng, China). After SDS-PAGE electrophoresis and membrane transfer, the total protein samples were incubated with antibody for immune reaction. The films were then sealed with 5% skim milk (containing 0.5% TBST) on the shaker for 1 hour. Then, the primary antibodies were added and shaken slowly at 4 °C overnight. The dilution of JNK antibody (GB12001, Wuhan Servicebio Technology Co., Ltd., Wuhan, China), p65 antibody (GB11142, Wuhan Servicebio Technology Co., Ltd. Wuhan, China) and β-actin antibody (24164-1-AP, Proteintech Group, Inc, Wuhan, China) were 1:2000, 1:1000 and 1:3000, respectively. Subsequently, the membrane was washed with TBST at room temperature for 3 times (5 min each time). After adding the secondary antibody (diluted 3000 times with TBST) and mixing with the membrane, incubate at room temperature for 30 min. After incubation, the membrane was washed with TBST at room temperature for 3 times (5 min each time). Finally, chemiluminescence was carried out in the dark room. The film was scanned and analyzed by the gel image processing system to obtain the molecular weight and optical density information of the protein.

### 2.7. Analysis of the Intestinal Microbiota Diversity

Analysis of the intestinal microbiota diversity was conducted according to the method described by Schneider et al. [28]. The analysis of microbial diversity in fecal samples was entrusted to GENEWIZ (Suzhou, China). The total DNA of microorganisms in fecal samples was extracted and qualified. The v3-v4 region of 16S rDNA in total DNA was amplified with 319F and 806R primers to construct a library for high-throughput sequencing. The same volume of 1 loading buffer (containing SYB green) was mixed with the PCR products. The mixture was quantified by 2% agarose gel electrophoresis and purified by using a QIAquick Gel extraction kit (Qiagen). Sequencing and data analysis were then carried out on an Illumina HiSeq platform by Novogene (Suzhou, China). After obtaining the original sequencing data, the sequencing information was analyzed to explore the diversity of fecal microorganisms. Further, 97% sequence similarity, which is roughly equivalent to the species level sequence difference in taxonomy, was used as OTU threshold. The obtained OTUs were classified based on the Silva (bacterial) taxonomy database. Alpha diversity of samples (Shannon index, rank abundance curve, Chao1 richness estimator, Ace richness estimator, Simpson diversity index) was analyzed using Mothur (version v.1.30) software.

### 2.8. Statistical Analysis

Data were expressed as means ± standard deviation (SD) for each group. SPSS software version 15.0 was used for all the data analysis. Statistical comparisons among multiple groups were performed using one-way ANOVA followed by the Bonferroni post hoc test. Two-way ANOVA followed by a Bonferroni post hoc test was used to analyze the weight data. Statistical analysis methods for microbiome are available upon request. *p* < 0.05 was considered to be statistically significant.

## 3. Results

### 3.1. Lactobacillus Plantrum MA2 Improved the Phenotype and Lipid Metabolism of the MCD-Induced NAFLD Rats

At the end of the experiment (56 days), compared with normal rats, rats on MCD diet had hair disorder, smaller body size and reduced food intake. These performances in the NAFLD group were consistent with the pathological characteristics of NAFLD. Figure 1a shows the body weights of rats on days 0, 7, 14, 21, 28, 35, 42, 49 and 56. Compared with the normal group, the weight of rats in the NAFLD group decreased significantly from the seventh day (*p* < 0.05). On the 56th day, the weight of rats in the NAFLD group was 21.6% lower than that of the normal group. The weight loss also indicated that the NAFLD rats were successfully constructed by the feeding of MCD diet. From Day 0 to Day 35, there was no significant difference in body weight between MA2 group and NAFLD group. On the 42th day of the experiment, the body weight of MA2 group began to be lower than that of NAFLD group (*p* < 0.05). The content of TG and TC in serum was detected to evaluate the lipid metabolism at the end of the experiment (56 days). Compared with the normal group, the TC and TG in the NAFLD group were significantly increased from 0.63 ± 0.06 mmol/L and 0.34 ± 0.18 mmol/L to 2.42 ± 1.21 mmol/L and 1.95 ± 0.65 mmol/L, respectively (Figure 1b). Further, in the MA2 group, the content of serum TC and TG were significantly lower than those of in the NAFLD group (0.64 ± 0.23 mmol/L and 0.25 ± 0.07 mmol/L, *p* < 0.05). The above results showed that although *L**. plantarum* MA2 did not restore the body weight of NAFLD rats, it significantly improved the lipid metabolism.

### 3.2. L. plantarum MA2 Improved Liver Lipid Deposition and Pathological Damage Caused by NAFLD

The appearance and histological structure of the livers in different groups were observed. As shown in Figure 2a, the liver in the normal group was bright red, hard and sharp. In contrast, the liver in the NAFLD group was round, blunt, dark red and greasy, showing the typical characteristics of fatty liver. Similar to the normal group, the liver in MA2 group was bright red, round, sharp edge, no greasy feeling or swelling. Then, we calculated the liver index of rats in different groups (Figure 2b). Compared with the normal group, the liver index of NAFLD group was significantly increased from 3.27 ± 0.38% to 4.73 ± 0.48% (*p* < 0.05). In contrast, the liver index in the MA2 group was 3.68 ± 0.31%, which was significantly lower than that of the NAFLD group (*p* < 0.05). Therefore, the appearance and index of liver in different groups showed that *L**. plantarum* MA2 feeding reduced liver fat accumulation in MCD-induced NAFLD rats.

Then, the histological structure of the livers in different groups of rats was observed after H&E staining (Figure 2c), and evaluated by NAFLD disease spectrum histological scoring system (Figure 2d). The results showed that the hepatocytes in the normal group were fat free, polygonal, dense, rich in cytoplasm, and with a central nucleus and clear lobular structure. In addition, no inflammatory cell infiltration or other pathological changes were observed in portal area of liver tissue. On the contrary, in the NAFLD group, the hepatocytes were obviously swollen, with vacuoles in the cytoplasm, and nuclear deviation or even nuclear disappearance caused by lipid droplets. The liver tissue was spread with different sizes of oil droplets and inflammatory cells. The analysis showed that compared with the normal group (each score was 0), the steatosis score, the lobular inflammation score and the ballooning score in NAFLD group reached 2.78 ± 0.44, 1.67 ± 0.5 and 2.00 ± 0, respectively (*p* < 0.05). The above results showed that the tissues of the liver in the NAFLD group were obviously damaged. In the MA2 group, the lipid droplet distribution, hepatocyte swelling, nuclear deviation and inflammatory cell infiltration in the liver were improved. The steatosis score (1.33 ± 0.5), the lobular inflammation score (1.22 ± 0.44) and ballooning score (1.11 ± 0.33) were significantly decreased (*p* < 0.05). The above results showed that the liver damage was partially relieved by *L**. plantarum MA2* intervention.

Subsequently, the damage of the liver tissue was further evaluated by the content of GPT and GOT in serum. The serum GPT and GOT content in the NAFLD group (195.87 ± 47.49 U/L and 156.09 ± 67.34 U/L) were significantly higher than those in the normal group (42.29 ± 10.88 U/L and 67.67 ± 15.79 U/L) (*p* < 0.05) (Figure 2e). In the MA2 group, the serum content of GPT (14.39.87 ± 10.77 U/L) and GOT (21.66 ± 18.79 U/L) were significantly reduced (*p* < 0.05). To sum up, the above histopathological analysis and serum index test results showed that supplementation of *L**. plantarum* MA2 significantly improved the pathological damage of the liver in MCD-induced NAFLD rats.

### 3.3. L. plantarum MA2 Protects the Rat Intestinal Mucosal Barrier System

Based on the importance of the intestinal barrier for liver inflammation in the “two hits” theory, we then tested the effect of *L*. *plantarum* MA2 on the intestinal barrier of NAFLD rats. First, H&E staining was used to observe the effect of strain MA2 on the physical barrier of the intestine. In the normal group, the small intestinal villi showed close arrangement, no gaps, and most of the mucosa was covered. In the NAFLD group, it was observed that the villi of the small intestine were sparsely arranged or even lost. Moreover, no mucosal coverage was observed in the NAFLD group (Figure 3a). For the MA2 group, although some villi on the top of the small intestine were lost, the length of villi on the bottom increased, and the mucosal coverage returned to normal (Figure 3a). The villi length was 770.2 ± 228.4 pixels in the NAFLD group which was significantly shorter than that in the normal group (389.2 ± 35.7 pixels, *p* < 0.05), while the value was significantly increased in the MA2 group (847.1 ± 347.9 pixels, *p* < 0.05). This indicates that *L*. *plantarum* MA2 protects the intestinal mucosal barrier from damage in the NAFLD state.

Subsequently, we tested the expression of claudin-1, a tight junction protein that plays an important role in maintaining the normal function of the intestinal epithelial barrier. As shown in Figure 3b, the RT-qPCR results showed that the relative expression of claudin-1 in the NAFLD group was significantly reduced compared with the normal group (0.23 ± 0.21, *p* < 0.05). This indicates that the intestinal barrier in the NAFLD group is severely damaged. In the MA2 group, the expression of claudin-1 was significantly increased (6.85 ± 0.76; *p* < 0.05, Figure 3b). The results indicated that *L**. plantarum* MA2 could protect the intestinal barrier by up-regulating the expression of claudin-1. What is more, the content of LPS in the serum of the rats can also confirm the effect of *L**. plantarum* MA2 on protecting the intestinal barrier. As shown in Figure 3c, compared with the normal group (134.98 ± 15.35 ng/L), the serum LPS level in the NAFLD group was significantly increased to 168.89 ± 17.45 ng/L. However, after intervention with *L**. plantarum* MA2 in NAFLD group (MA2 group), the data decreased significantly to 128.01 ± 10.22 ng/L (Figure 3c).

### 3.4. Lactobacillus Plantrum MA2 has a Significant Regulating Effect on the Intestinal Microbiota

Since the intestine is an important target for probiotics, we used high-throughput sequencing to analyze the composition of the intestinal microbiota in different groups of rats. In terms of intestinal microbiota diversity, our research shows that feeding *L**. plantrum* MA2 prevented the altered α diversity in the intestinal microbiota of NAFLD mice (Table 2). In the NAFLD group, the Ace and Chao1 indexes (264.93 ± 9.48 and 273.31 ± 8.94) were significantly lower than that of the normal group (293.11 ± 12.74 and 293.29 ± 14.05) (*p* < 0.05). However, in the MA2 group, the decreased Ace and Chao1 indexes were significantly higher than those in the NAFLD group, which were 290.33 ± 5.2 and 292.78 ± 6.93 (*p* < 0.05), respectively. This result indicates that *L**. plantrum* MA2 restored the total number and abundance of rat intestinal species that were reduced in the MCD diet induced group. As for the Shannon index and Simpson index, which represent species diversity, analysis shows that the changes between the three groups are not significant. The coverage rate of the sequencing sample library Good’s coverage reached one, indicating that the data of α diversity is highly reliable. The above results indicate that the total number and abundance of the intestinal microbiota of rats with NAFLD disease constructed in this study are significantly reduced, but the species diversity changes little. Moreover, through the intervention of *L**. plantrum* MA2, the species richness reduced in the rats with NAFLD disease was significantly restored.

Subsequently, the species information of the rat intestinal microbiota was annotated and analyzed at the phylum level and genus level. At the phylum level (Figure 4a), taxonomic profiling data demonstrated that the NAFLD lead to a significant decrease in *Firmicutes* to *Bacteroidetes* ratio and an increase in *Proteobacteria*. According to the data of the MA2 group, the treatment of *L**. plantrum* MA2 increased the ratio of *Firmicutes* to *Bacteroidetes* and decreased the abundance of *Proteobacteria*, which made the composition of the intestinal microbiota similar to that of the normal group. The excessive Gram-negative bacteria in *Bacteroidetes* phylum may increase the LPS content in the intestine or even the circulatory system, and many *Proteobacteria* are opportunistic pathogens in many cases [29]. For the results at the genus level (Figure 4b), the dominant genus of the intestinal microbiota of rats in the normal group were *Lactobacillus* and *Ruminococcaceae* (UCG-014&UCG-005). In the NAFLD group, the dominant genus were changed to *Lachnospiraceae* (unclassified genus), *Escherichia-Shigella*, *Bacteroides*, and [*Eubacterium*]_*coprostanoligenes*_group and *Blautia*. In the MA2 group of rats, the dominant genus of the intestinal microbiota were restored to *Lactobacillus*, *Ruminococcaceae* and [*Eubacterium*]_*coprostanoligenes* group similar to the normal group. At the same time, the content of the dominant genera in the NAFLD group, such as *Lahnospiraceae*, *Escherichia-Shigella*, *Bacteroides* and *Blautia*, significantly reduced (Figure 4). According to the above changes, we concluded that the bacterial groups that cause obesity and inflammation in the intestines of the NAFLD group rats increased significantly, while the content of probiotics decreased significantly. The intervention of *L**. plantarum* MA2 significantly improved the intestinal microecology of rats, increased the number of beneficial bacteria and reduced the number of harmful bacteria, and had a restorative effect on the disordered intestinal microbiota in the NAFLD rats.

### 3.5. L. plantarum MA2 Reduces Liver Inflammation by Down-Regulating the Expression of Inflammation-Related Pathway Proteins

Based on the improvement effect of *L**. plantarum* MA2 on the intestinal barrier and intestinal microbiota, the transcription levels of TLR4-MYD88 and related genes in the downstream inflammation signaling pathways that are closely related to the pathogenesis of NAFLD in the rat liver were detected. The results showed that the relative transcription level of TLR4 in the NAFLD group (4.84 ± 2.74) was significantly higher than that in the normal group, while the relative transcription level of TLR4 in the liver of the MA2 group was significantly reduced (1.64 ± 0.68; Figure 5a). The expression of MyD88 downstream of TLR-4 also had a similar trend. Compared with the normal group, the relative transcription level of MyD88 in the NAFLD group was significantly increased to 18.13 ± 2.74. In the MA2 group, the relative transcription level of MyD88 was significantly lower than that of the NAFLD group (2.6 ± 0.92; Figure 5b). The above results indicate that in the NAFLD group, the invasion of LPS up-regulated the transcription of TLR4 in the liver, which in turn led to the significant activation of MyD88. However, feeding of the probiotic *L**. plantarum* MA2 significantly inhibited the activation. Subsequently, the regulatory effects of *L**. plantarum* MA2 on the NF-κB signaling pathway and JNK (c-Jun N-terminal kinase) signaling pathway downstream of the TLR-4-Myd88 pathway were tested. For the NF-κB signal pathway, the expression of p65 protein was detected. The results showed that compared with the normal group (0.5 ± 0.06), the expression of p65 protein in the NAFLD group (1.33 ± 0.20) was significantly increased by 2.66 times (*p* < 0.05). The expression of related proteins in the MA2 group (0.84 ± 0.04) was significantly reduced (*p* < 0.05), only 63.16% of the NAFLD group (Figure 5c). For the JNK signaling pathway, the phosphorylation of JNK kinase and the relative transcription level of its downstream AP-1 were detected. As shown in Figure 5d, compared with the normal group (0.23 ± 0.03), the JNK kinase activity of the NAFLD group was significantly increased to 1.47 ± 0.08 (*p* < 0.05). In the MA2 group, JNK kinase activity was significantly reduced to 0.5 ± 0.024 (*p* < 0.05) compared with the NAFLD group. As for the relative transcription level of AP-1, it was significantly higher in the NAFLD group, which was 25.32 ± 3.42 times that of the normal group. In the MA2 group, it was only 3.57 ± 0.75 times that of the normal group, which was significantly lower than the NAFLD group (Figure 5e). All the above results indicate that *L**. plantarum* MA2 can alleviate the excessive activation of liver inflammation in NAFLD rats by significantly inhibiting the expression of TLR4-MYD88 and its downstream pathways.

We also tested the relative expression levels of inflammatory factors (TNF-α, IL-1β, IL-6) and anti-inflammatory factors (IL-4 and IL-10) in the liver. For the inflammatory factors, as shown in Figure 6a, the relative expression of inflammatory factors TNF-α, IL-1β and IL-6 in NAFLD group increased significantly, reaching 5.3 ± 1.87 times, 53.1 ± 10.89 times and 32.45 ± 13.8 times of the normal group, respectively. In the MA2 group, their relative expression were significantly down-regulated to 1.32 ± 0.35 times (TNF-α), 12.89 ± 3.66 times (IL-1β) and 9.24 ± 0.83 times (IL-6) of the normal group, respectively. As for the anti-inflammatory factors IL-4 and IL-10 (Figure 6b), compared with the normal group, their relative expression in the NAFLD group were significantly increased to 1.8 ± 0.34 and 2.01 ± 0.84 times. In the MA2 group, the relative expression of IL-4 and IL-10 have a more significant increase that reached 8.93 ± 4.96 times and 3.36 ± 0.52 times of the normal group, respectively. Therefore, it can be concluded that oral administration of *L**. plantarum* MA2 could down-regulate the expression of inflammatory factors and promote the expression of anti-inflammatory factors in the liver of MCD diet induced NAFLD rats, thereby alleviating the inflammatory in the liver.

## 4. Discussion

### 4.1. L. plantrum MA2 Can Improve Lipid Metabolism and Relieve Liver Fat Accumulation and Liver Pathological Damage in MCD-Induced NAFLD Rats

NAFLD is a high-incidence liver disease characterized by the accumulation of liver triglycerides (TG) exceeding 5% of the total liver weight [30]. Due to the lack of methionine in the MCD diet, it will reduce the methyl groups required for choline synthesis, thereby reducing the synthesis of lipoproteins in rats. The MCD-induced NAFLD model would exhibits symptoms of lipid metabolism disorders, deposited triglycerides, hyperlipidemia and weight loss.

In this study, as expected, the NAFLD group showed symptoms such as weight loss. The body weight loss could not be restored by feeding the strain *L*. *plantrum* MA2. However, the Lactobacillus plantrum MA2 intervention in the NAFLD rats could significantly reduce the TG and TC content (*p* < 0.05) in the serum. The recovery effect indicates that *Lactobacillus plantrum* MA2 has an improved effect on lipid metabolism and can relieve the symptoms of NAFLD to a certain extent [31,32]. H&E staining of the liver showed that MCD-fed rats showed obvious fat accumulation and denaturation (steatosis), which is an important feature of NAFLD. In addition, hepatic steatosis is also the basis of NASH, NASH-related cirrhosis and hepatocellular carcinoma (HCC) [33,34]. Therefore, we focused on observing whether *L*. *plantrum* MA2 can improve this kind of steatosis. The results of H&E staining showed that compared with the NAFLD group, in the MA2 group, the fat deposition in the liver tissue was significantly reduced, and the hepatic steatosis or even the inflammatory cell infiltration were also significantly improved. At the same time, the content of GPT and GOT, which are considered to be important markers for estimating the degree of liver damage [35], were observed. The levels of GTP and GOT in the MA2 group were significantly reduced, indicating that the intervention of *L*. *plantarum* MA2 can reduce MCD-induced liver damage and protect liver function. These data indicate that *L*. *plantarum* MA2 improved the liver phenotype of NAFLD rats and alleviated liver damage.

### 4.2. L. plantarum MA2 Reduces Liver Inflammation by Acting on the LPS/TLR4 Inflammatory Pathway

Many studies have shown that the LPS/TLR4 signaling pathway mediates inflammation, oxidative stress, insulin resistance and liver fibrosis, and is one of the key factors in the pathology of NAFLD [36,37]. In the liver, TLR4 specifically recognizes LPS, and then activates intracellular signal transduction through MyD88, thereby inducing an immune response. Excessive LPS signaling may further activate the downstream NF-κB signaling pathway through the increase of T lymphocytes and B lymphocytes caused by the activation of the TLR4-MyD88 signaling pathway [38]. The activation of the NF-κB signaling pathway leads to the phosphorylation of p65 in the nucleus, which then triggers a series of physiological or pathological effects (such as the production of inflammatory cytokines, including TNF-α, IL-1β and IL-6), which eventually leads to hepatitis. The JNK signaling pathway is an important branch of the MAPK (mitogen-activated protein kinase) pathway, which plays an important role in many physiological and pathological processes (such as cell cycle, reproduction, apoptosis and cell stress). Activator protein 1 (AP-1) is located downstream of the JNK signaling pathway and is an important nuclear transcription factor that can regulate gene expression in response to various stimuli (including cell differentiation, proliferation and apoptosis).

Our research also shows that *L**. plantarum* MA2 inhibits the activity of NF-κB p65 protein, resulting in a decrease in the relative expression of inflammatory cytokines TNF-α, IL-1β and IL-6. In addition, *L**. plantarum* MA2 also causes the activation of T cells, increases the relative expression of anti-inflammatory factors IL-4 and IL-10, and inhibits the secretion of pro-inflammatory cytokines induced by LPS. Moreover, *L**. plantarum* MA2 significantly reduced the increase in the relative expression levels of JNK and AP-1 in the liver of NAFLD rats induced by MCD. Down-regulation of AP-1 protein transcription level inhibits the relative expression of its downstream pathways, thereby reducing apoptosis, cell lipid accumulation, and pathological changes in hepatocytes.

### 4.3. L. plantarum MA2 Works by Improving the Structure of the Intestinal Microbiota and the Barrier of the Intestinal Mucosa

The intestine is very important to human health, and its role in nutrient absorption, fat metabolism and weight regulation has been widely recognized. In the “gut liver axis” theory, the intestinal microbiota has a close influence on the liver by affecting the metabolism of bile acids (BAs), LPS and SCFA [39]. Intestinal microbiota plays an important role in the pathogenesis of NAFLD through a variety of modes, including the regulation of intestinal metabolites and intestinal mucosal permeability. Under normal physiological conditions, the intestine is the largest reservoir and endotoxin reservoir in the human body, and changes in the intestinal microbiota are most likely to cause a “second hit”.

Previous studies have shown that one of the main reasons for the increase in serum LPS is the increase in the proportion of intestinal Gram-negative bacteria [40,41,42]. Our experimental results also indicate that this “second hit” event triggered by intestinal-derived LPS may occur in our rat model. Our study also observed a significant decrease in the probiotic *Lactobacillus* content in the NAFLD group, while the *Bacteroides* and *Escherichia-Shigella* content that may increase the intestinal LPS content significantly increased. Many studies have shown that *L. plantarum* administration also inhibited pathogenic bacterial growth, while promoting growth of probiotics such as *Allobaculum*, *Lactobacillus*, and, most markedly, *Bifidobacterium* and corrects gut microbiota disorders caused by a high-fat and fructose diet. Moreover, *L. plantarum* treatment of NAFLD mice improved intestinal barrier integrity and attenuated high-fat and fructose diet (HFD/F)-induced inflammation [43]. In our study, *L**. plantarum* MA2 also played a similar role as described above by regulating the intestinal microbiota. This regulation includes changes in the alpha diversity of the intestinal microbiota, restoring the ratio of beneficial bacteria in NAFLD, and reducing the ratio of harmful bacteria in NAFLD. This regulation effect includes the increase in the abundance of the intestinal microbiota alpha diversity, the restoration of the reduced proportion of beneficial bacteria (such as, *Lactobacillus* and *Ruminococcaceae*) and the increased proportion of harmful bacteria (such as, *Lachnospiraceae, Escherichia−Shigella, Bacteroides* and *Blautia*).

In addition, the effect of strains on the barrier function of the intestinal mucosa will also affect the progression of NAFLD. Goblet cells (GCs) are specialized epithelial cells that line multiple mucosal surfaces and have a well-appreciated role in barrier maintenance through the secretion of mucus. GCs can separate the materials in cavities from the intestinal epithelium and prevent the invasion of pathogenic microorganisms in various ways. LPS can induce morphological changes in goblet cells and abnormal expression of mucins [44]. Under normal physiological conditions, a small amount of LPS will cross the normal intestinal barrier and enter the liver [45], and then be recognized by pattern recognition receptors (including TLR) on Kupffer cells and then eliminated [46,47]. LPS induces the hepatic stellate cells to express higher levels of lipopolysaccharide receptors. The high level of LPS triggers the “second hit” event, which stimulates a strong inflammatory response and causes liver damage, accelerating the development of NAFLD [48]. Intestinal epithelial mucosal tight junction protein is an important protein of the intestinal barrier, and its high expression level can prevent excessive LPS from entering the intestinal wall and blood stream [49,50]. Therefore, the expression level of tight junction protein is an important indicator for evaluating intestinal permeability. This effect possibly stemmed from *L. plantarum* promotion of SCFA-producing bacterial growth. SCFA are known to exert multiple beneficial effects on the host, including promotion of lactic acid bacterial growth, reduction of insulin resistance and inflammation, improvement of intestinal barrier function, and stimulation of lipid oxidation [43]. Our results indicate that *L**. plantarum* MA2 intervention can increase the expression of tight junction proteins occludin-1 and claudin-1. Therefore, this improvement effect on the intestinal mucosal barrier may have caused a significant decrease in the blood LPS level in rats.

## 5. Conclusions

In this study, we demonstrated that the probiotic *L**. plantarum* MA2 can interfere with NAFLD through the intestinal microbiota and intestinal mucosal barrier. *L**. plantarum* MA2 can reduce the growth of harmful bacteria, restore the damage of the intestinal barrier caused by LPS, thereby reducing the entry of LPS from the intestine to the liver, and inhibit the activation of LPS-TLR4 and downstream inflammatory signaling pathways. Therefore, our preclinical studies on animal models show that oral probiotic *L**. plantarum* MA2 has the potential possibility to become a treatment option for NAFLD. Our research has opened up a new way to develop effective treatments for NAFLD based on intestinal microbiota.

## Figures and Tables

**Figure 1 foods-10-03126-f001:**
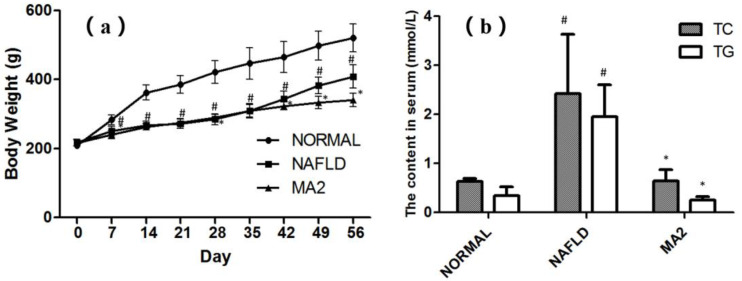
Effects of *Lactobacillus plantrum* MA2 on body weight and lipid metabolism in NAFLD rats. (**a**) Boday weight of the rats in different group from Day 0 to 56, (**b**) the content of TC and TG in serum at Day 56, (# indicates *p* < 0.05 compared with the normal group; * indicates *p* < 0.05 compared with the NAFLD group. NORMAL: normal group, NAFLD: non-alcoholic fatty liver disease, MA2: *L. plantarum* MA2 treatment group).

**Figure 2 foods-10-03126-f002:**
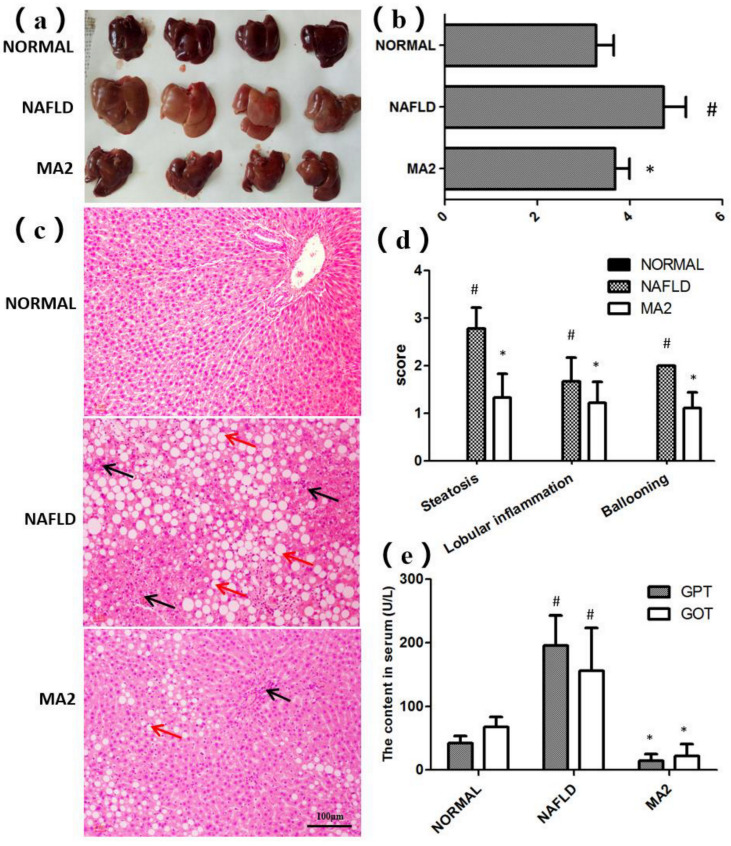
*L**. plantarum* MA2 prevented liver fat accumulation and pathological damage in NAFLD rats. (**a**) Liver appearance, (**b**) liver index, (**c**) morphology of liver tissues (H&E staining). H&E-stained sections revealed macrovesicular and microvesicular steatosis throughout the entire lobules (red arrows), as well as scattered lobular and perivenular inflammation (black arrows). (Original magnification: ×200; Bars = 100 μm). (**d**) Liver histopathology score, (**e**) GPT and GOT activity in serum. The values were presented as mean ± standard error of mean (*n* = 8), (# indicates *p* < 0.05 compared with the normal group; * indicates *p* < 0.05 compared with the NAFLD group. NORMAL: normal group, NAFLD: non-alcoholic fatty liver disease, MA2: *L**. plantarum* MA2 treatment group).

**Figure 3 foods-10-03126-f003:**
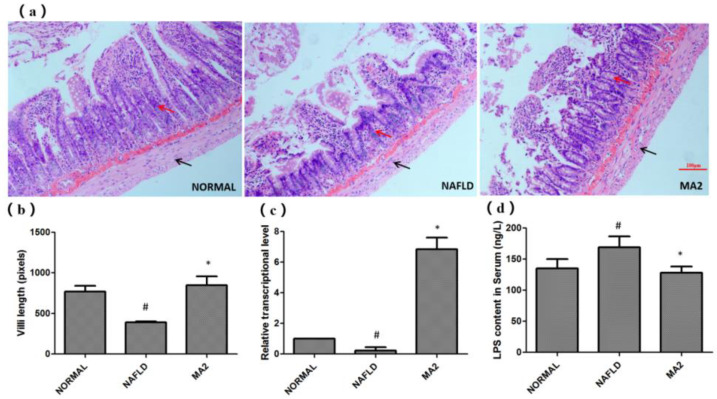
The protective effect of *L**. plantarum* MA2 on the intestinal barrier of rats. (**a**) The tissue morphology of the ileum in different groups (H&E staining, scale bars: 50 mm), the villi and the muscula are marked by red and black arrows, respectively, (**b**) the villi length, in pixels, *n* = 10, (**c**) expression of intestinal tight junction protein claudin-1, (**d**) serum LPS levels in different groups of rats, (# indicates *p* < 0.05 compared with the normal group; * indicates *p* < 0.05 compared with the NAFLD group. NORMAL: normal group, NAFLD: non-alcoholic fatty liver disease, MA2: *L**. plantarum* MA2 treatment group).

**Figure 4 foods-10-03126-f004:**
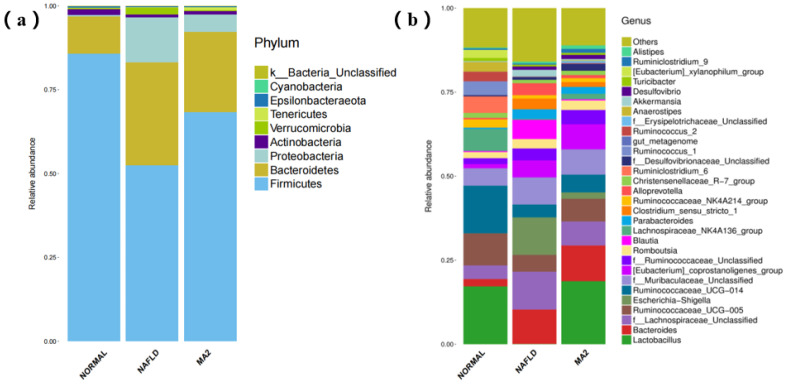
The distribution of the intestinal microbiota in different groups of rats. (**a**) At the phylum level, (**b**) at the genus level. (NORMAL: normal group, NAFLD: non-alcoholic fatty liver disease, MA2: *L**. plantarum* MA2 treatment group).

**Figure 5 foods-10-03126-f005:**
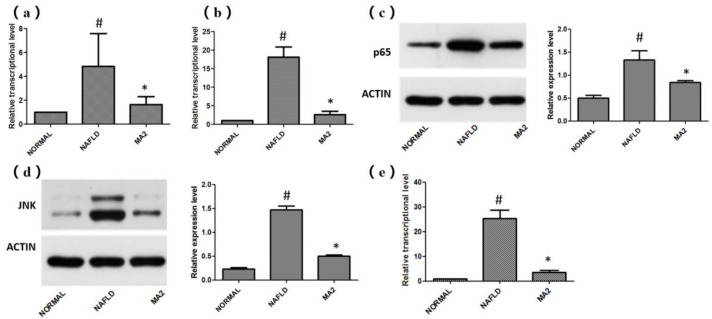
Effects of *L**. plantarum* MA2 on the expression levels of related genes in rat liver LR4-MYD88 and its downstream pathways. (**a**) Relative transcription levels of TLR4 mRNA, (**b**) relative transcription levels of MyD88 mRNA, (**c**) the expression level of p65 protein, (**d**) JNK kinase activity, (**e**) relative transcription level of AP-1 mRNA. (# indicates *p* < 0.05 compared with the normal group; * indicates *p* < 0.05 compared with the NAFLD group. NORMAL: normal group, NAFLD: non-alcoholic fatty liver disease, MA2: *L. plantarum* MA2 treatment group).

**Figure 6 foods-10-03126-f006:**
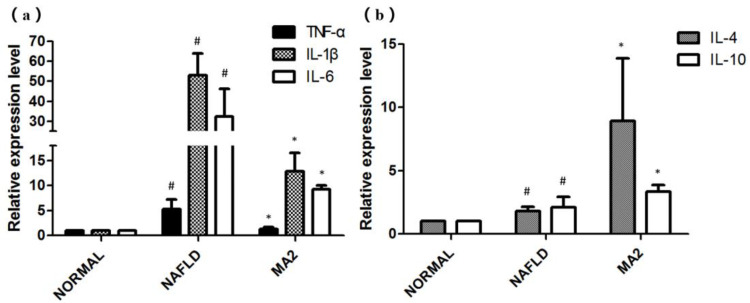
The influence of *L**. plantarum* MA2 on the relative expression levels of inflammatory factors in rats. (**a**)TNF-α, IL-1βand IL-6 and anti-inflammatory factors; (**b**) IL-4 and IL-10. (# indicates *p* < 0.05 compared with the normal group; * indicates *p* < 0.05 compared with the NAFLD group. NORMAL: normal group, NAFLD: non-alcoholic fatty liver disease, MA2: *L. plantarum* MA2 treatment group).

**Table 1 foods-10-03126-t001:** The primer sequences and their target genes.

Target Genes	Forward (5’–3’)	Reverse (5’–3’)
TNF-α	GACGTGGAACTGGCAGAAGAG	TCTGGAAGCCCCCCATCT
IL-6	AGAGGAGACTTCACAGAGGATACC	AATCAGAATTGCCATTGCACAAC
IL-1β	CTGTGTCTTTCCCGTGGACC	CAGCTCATATGGGTCCGACA
IL-10	AGGGCACCCAGTCTGAGAACA	CGGCCTTGCTCTTGTTTTCAC
MyD88	TGCGTCTGGTCCATTGCT	TCACATTCCTTGCTTTGC
TLR4	CATGAGCGCTGAAGTGGTGA	CGATCGATAATGGTGAGACC
Claudin-1	AAAGTGAAGAAGGCCCGTATA	TAATGTTGGTAGGGATCAAAGG
AP-1	CTGAAGGGATTGGAGAC	TGGGAGCGACATAGGA
IL-4	GCTATTGATGGGTCTCACCC	CAGGACGTCAAGGTACAGGA
β-actin	TGGGACGATATGGAGAAGAT	ATTGCCGATAGTGATGACCT

**Table 2 foods-10-03126-t002:** α Diversity index in different group of rats.

Group	Ace	Chao1	Shannon	Simpson	Good’s Coverage
Normal	293.11 ± 12.74	293.29 ± 14.05	5.85 ± 1.01	0.95 ± 0.042	1
NAFLD	264.93 ± 9.48 #	273.31 ± 8.94 #	5.66 ± 0.7	0.94 ± 0.053	1
MA2	290.33 ± 5.2 *	292.78 ± 6.93 *	5.73 ± 0.49	0.94 ± 0.034	1

# Indicates *p* < 0.05 compared with the normal group; * indicates *p* < 0.05 compared with the NAFLD group. NORMAL: normal group, NAFLD: non-alcoholic fatty liver disease, MA2: *L**. plantarum* MA2 treatment group.

## Data Availability

The data presented in this study are available on request from the corresponding author.

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
