# Peer review of "Lactobacillus plantarum MA2 Ameliorates Methionine and Choline-Deficient Diet Induced Non-Alcoholic Fatty Liver Disease in Rats by Improving the Intestinal Microecology and Mucosal Barrier"

_foods, 2021, doi:10.3390/foods10123126_

Round 1
Reviewer 1 Report
The authors presented interesting research of influence of the probiotic 65 Lactobacillus plantarum MA2 on the MCD-induced NAFLD.
The manuscript is presented in a well-structured manner, but there are some issues that need to be clarified:
Introduction:
- Please provide more relevant citation explaining how microbiota plays 51 an important role in the pathogenesis of human liver diseases, especially in NAFLD. Authors provide only two citations…
Materials and Methods:
- Animal Treatment and sample collection :
Line 81 – “…. rats were fed with methionine choline 86 deficiency (MCD) diet for 8 weeks to induce NAFLD diseases…” – only NAFLD group was fed 8 weeks, all groups were fed 8 weeks?
Line 101 – “At the end of the experiment…..” When …after 8 weeks?
Please, write down how long the experiment lasted.
- RNA extraction and real-time PCR
Please explain the method you used to calculate the fold change in PCR reaction.
Did you use housekeeping genes? Please provide which housekeeping genes you used and add them into Table 1.
- Detection of protein expression
Please add the primary antibody used in this study and dilutions for each of them.
- Statistical analysis
Line 168 - ….”One-way analysis of variance 168 (ANOVA) was used to analysis the differences between the groups p<0.05 ….” In text, there are also p< 0.01…
Did you use any of post-hoc tests? How you find the statistical differences between groups?
p-values should be written in the same way throughout the manuscript – p < 0.05
Results
Figure 5. c and d – There was no statistical differences between groups?
Reviewer 2 Report
Dear Authors
After analysis of the manuscript named "Lactobacillus plantarum MA2 ameliorates methionine and choline-deficient diet induced non-alcoholic fatty liver disease in rats by improving the intestinal microecology and mucosal barrier", I consider that the same gains merit, relevance for this journal . However, special attention is needed regarding the methodology/morphological results of the liver and small intestine.
Review (points to be analyzed)
1.Abstract-NAFLD: Non-alcoholic fatty liver disease; in the rest of the text: Nonalcoholic fatty liver disease. Suggestion: standardize
2.Materials and Method
- Animal treatment and sample collection – which small intestine samples were fixed? Duodenum, jejunum or ileum (or initial, middle or final segment). This definition is absolutely necessary in histological and microbiota analyses.
- Histological observation and evaluation – The number of liver samples per group is extremely reduced (only three); I also suggest more detailing of the methodology (cut thickness, equipment used, number of images used in liver analysis); the use of the Sudan III histochemical method for evidencing lipids would be more adequate to characterize the condition of hepatic steatosis.
- I also suggest that this topic includes the gut methodology part.
3.Results
- How was reduced exercise ability rated? Detail
- The direct statement that there was a reduction in the accumulation of fat in the liver from HE stained images may not be the most indicated.
- Figure 2: mention in the caption what rats are, which dye was used and indicate in the photo the characteristic vacuolization and lobular inflammation.
- Figure 3 (a): I suggest that the authors review the histological images presented in terms of quality, resolution and preservation of the material, in addition to identifying the structures and informing the dye. Suggestion: present only the intestinal mucosa.
- As morphometry of the intestinal mucosa was not performed, the assertion of increased villi length is not adequate (line 256).
- Replace the name intestinal flora with intestinal microbiota.
- Figure 6: include rats
4.Discussion
- Review lines 409-410 (4.1)
- Suggestion: Evaluate or discuss as possible the role of goblet cells within the response obtained due to their importance in the formation of the intestinal barrier.
Kind regards,
Author Response
Please see the attachmen

Reviewer 3 Report
Title: Lactobacillus plantarum MA2 ameliorates methionine and choline-deficient diet induced non-alcoholic fatty liver disease in rats by improving the intestinal microecology and mucosal barrier
In this article, the authors investigated the potential application of Lactobacillus plantarum MA2 in alleviating Non-alcoholic fatty liver disease. The authors used the methionine and choline-deficient (MCD) diet-induced NAFLD rat model to address this. The topic is important but there are a few concerns
Comments
- The major issue is it is not a temporal study with rats sacrificed at frequent intervals to claim the sequence of events in a particular order. This needs to be corrected throughout.
- Lactobacillus plantarum MA2 preparation lacks details – what is the drying method and how many are present after drying? Are you gavaging 1x108 cfu/gram body weight of mice? What is the medium of gavage? etc
- Bacterial OTUs – are they open or closed references and why did they choose one rather than another?
- How is the histopathology score calculated?
- Indicate in the figures what you are talking about liver and intestine morphology pictures – put the arrows and specify the differences
- Write up of the needs improvement –incomplete sentences, spelling mistakes and spacings especially before and after signs. For example line 174 – “At the end of experimental (56 days),” You mean experimental period? Line 146 The film were – I think the films were, the spacings before and after ±

Reviewer 4 Report
This paper discusses the findings of a controlled animal study investigating the effects of Lactobacillus plantarum MA2 in NAFLD markers, and show promising results for MA2 supplementation in NAFLD markers.
Overall the paper needs to address some key points to improve the validity/novelty of the study. There is a need for a rational to justify this study and its novelty, given the large body of evidence and systematic reviews/meta-analysis concluding that probiotic supplementation is beneficial to NAFLD. Unclear what this specific study done in mice adds compared to what is known in human clinical trials. Currently the results section is a bit difficult to follow as it is too long and includes discussion rather than reporting only, and it is unclear when results refer to change from baseline as this is often only presented for NAFLD group, not MA2 group. The wording here should be consistent with what presented, e.g. the use of 'improved' and similar wording needs to be accompanied by before and after data or by a change from baseline measurements.
Specific comments:
Abstract
Please include numerical results in this section when discussing changes in outcome measurements (i.e. improvement, by how much in which marker). Example: "intestinal mucosal barrier and the the structure of intestinal 16 flora were also improved." Included pre-post changes in the biological marker for mucosal barrier and p value of this.
Introduction
Line 47-49. Please include reference for 5% weight loss goal as treatment and poor compliance with this approach.
Lines 50-62: When discussing previous studies, please indicate duration of intervention, probiotic dose and type of participants. This sections could include more relevant studies to justify the need of this study done in rats rather than humans, since there are a few human trials done with probiotics.
E.g.: https://academic.oup.com/ajcn/article/110/1/139/5498099?login=true
https://academic.oup.com/nutritionreviews/article/76/11/822/5065715?login=true
- End of introduction, should state the aims of the paper/study and hypothesis, rather than the conclusion of the current study.
- Introduction needs to provide a rational for choosing a specific probiotic in the current study (Lactobacillus plantarum MA2). No previous studies on this strain have been presented in the introduction.
Methods
Lines 80-90: Indicate reference/s to justify the selection of methionine choline deficiency (MCD) diet for NAFLD disease, including duration of diet and type of mice.
Lines 99-100: why were samples collected only from a sub-group of animals and not all of them to allow for a longitudinal comparison (repeated measures).
Sections 2.3-2.5-2.6 -2.7 need references to support the methodology used.
Section 2.7 needs more detail in the bioinformatic analysis.
Statistical analysis: why was one-way anova used instead of 2-way (time x group). more detail needed here on what was the primary outcome and whether analysis was adjusted for any confounders, or data normalised (and how).
Results
Figure 1B: please use more distinct colours in each group, a currently they are very similar and difficult to distinguish. Indicate in legend the time-point of data ( is this after 56 days of supplementation?)
Lines 206-207 "NAFLD group was significantly increased from 3.27±0.38% to 206 4.73±0.48ï¼…. In contrast, the liver index in the MA2 group was 3.68±0.31%". Please indicate p value and the change in MA2 group, as currently only post-intervention values are presented.
Lines 229-231: like the previous comment, please indicate before/after values or change from baseline value when reporting things like "improvement". Same comment applies for lines 235-240
Figures: please include in all legends if the data presented is from post-intervention or if it is change from baseline values.
lines 283-284: "feeding Lactobacillus plantrum 282 MA2 restored the altered α diversity in the intestinal flora of NAFLD mice" this sentence implies that NAFLD group was treated with probiotics after the disease manifested, not along side. Please amend for clarity.
Section 3.5 - Some text here refers to figure 4d and 4e but these panels are not present in figure 4. Was this meant to be referring to figure 5? Please amend. In this section, some results are discussed as 'significant' but in the respective figure these are not significant - i.e. lack of * and # in figures 4c and 4d. Please amend accordingly.
Discussion
Section 4.1 - first paragraph belongs to Methods, also in line with my previous comment on that section (references/rational for study design).
lines 411-412: unclear sentence
Section 4.2-4.3 : please remove excessive background information and limit section to discuss current results vs previous research, and what this means.
Section 4.3: here the role of L plantarum should be discussed - why this specific bacteria is beneficial for NAFLD and how does it affects gut diversity results that were observed here?
How does L plantarun affect gut integrity and expression of tight function proteins? How do we know is this specific bacteria and not others present in the gut?
Round 2
Reviewer 1 Report
The Authors have provided answers to all comments and made corrections according to suggestions.
Reviewer 4 Report
Thank you for addressing the comments, the manuscript and research is much clear now.